# Cholesterol 25-Hydroxylase Suppresses Swine Acute Diarrhea Syndrome Coronavirus Infection by Blocking Spike Protein-Mediated Membrane Fusion

**DOI:** 10.3390/v15122406

**Published:** 2023-12-11

**Authors:** Dakai Liu, Da Shi, Hongyan Shi, Liaoyuan Zhang, Jiyu Zhang, Miaomiao Zeng, Tingshuai Feng, Xiaoman Yang, Xin Zhang, Jianfei Chen, Zhaoyang Jing, Zhaoyang Ji, Jialin Zhang, Li Feng

**Affiliations:** State Key Laboratory for Animal Disease Control and Prevention, Harbin Veterinary Research Institute, Chinese Academy of Agricultural Sciences, Xiangfang District, Haping Road 678, Harbin 150069, China; liudakai0404@163.com (D.L.); shy2005y@163.com (H.S.); zhangliaoyuanzhu@foxmail.com (L.Z.); zhangjiyu0429@163.com (J.Z.); 18790286972@163.com (M.Z.); fengtingshuai@126.com (T.F.); xiaomanyang9766@163.com (X.Y.); zhangxin2410@163.com (X.Z.); chenjianfei@126.com (J.C.); 15204604415@163.com (Z.J.); zy_ji2010@163.com (Z.J.); zhangjialin0106@gmail.com (J.Z.)

**Keywords:** cholesterol 25-hydroxylase, SADS-CoV, spike protein, membrane fusion

## Abstract

Swine acute diarrhea syndrome coronavirus (SADS-CoV) is an emerging porcine intestinal coronavirus that can cause acute diarrhea, vomiting, rapid weight loss, and high mortality in newborn piglets. Cholesterol 25-hydroxylase (CH25H) is a molecular mediator of innate antiviral immunity and converts cholesterol to 25-hydroxycholesterol (25HC). Previous studies have reported that CH25H and 25HC have an antiviral effect against multiple viruses. However, the interplay between SADS-CoV infection and CH25H or 25HC is still uncertain. Here, we found that CH25H and its enzymatic product 25HC restrained SADS-CoV replication by blocking membrane fusion. Our results show that CH25H was upregulated by SADS-CoV infection in vitro and in vivo, and that it was an IFN-stimulated gene in porcine ileum epithelial cells. Moreover, CH25H and CH25H mutants lacking catalytic activity can inhibit SADS-CoV replication. Furthermore, 25HC significantly suppressed SADS-CoV infection by inhibiting virus entry. Notably, we confirmed that CH25H and 25HC blocked SADS-CoV spike protein-mediated membrane fusion. Our data provide a possible antiviral therapy against SADS-CoV and other conceivable emerging coronaviruses in the future.

## 1. Introduction

Coronaviruses (CoVs) are enveloped, single-stranded, positive-sense RNA viruses that can infect humans and a wide range of animals and cause diseases of varying severity, including respiratory, enteric, and neurological diseases [1]. The subfamily *Orthocoronavirinae* of the family *Coronaviridae* is currently classified into four genera: *Alpha-*, *Beta-*, *Gamma-*, and *Delta-coronavirus*. Over the past few decades, pigs have suffered from severe CoV diseases with significant economic impacts [2]. Before 2016, five porcine CoVs were identified, including transmissible gastroenteritis virus and its variant porcine respiratory virus, porcine epidemic diarrhea virus (PEDV), porcine hemagglutinating encephalomyelitis virus, and porcine deltacoronavirus (PDCoV). In 2017, swine acute diarrhea syndrome coronavirus (SADS-CoV) was the sixth porcine CoV discovered [3]. SADS-CoV can cause swine acute diarrhea syndrome in piglets. Its clinical outcomes include acute diarrhea, vomiting, rapid weight loss in piglets <1 week old, and high mortality (>90%) [4]. Currently, no vaccine or antiviral drug against SADS-CoV is available [5]. SADS-CoV belongs to the genus Alphacoronavirus of the family Coronaviridae. Genomic sequence analysis demonstrated that SADS-CoV shared > 90% sequence similarity with Rhinolophus bat coronavirus HKU2, suggesting that SADS-CoV was the first bat-derived CoV that spilled over from bats to infect pigs directly [6]. It is reported that SADS-CoV infects cell lines from several species, including pigs, rodents, humans, chickens, and bats [7]. In vivo, evidence confirmed that in addition to pigs, mice and chickens were also susceptible to SADS-CoV infection [8]. Although the structure and composition of viruses are simple, the process of their invasion into host cells is very complex [9]. Coronaviruses can enter cells via fusion either directly at the cell surface or can be internalized through the endosomal compartment [10]. The entry of coronaviruses relies on a specific interaction between the spike (S) protein on the virion surface and a host cell receptor [11]. There is a large ectodomain and a very short endodomain on the S protein. The ectodomain consists of two domains: the N-terminal receptor-binding domain S1, which is responsible for receptor binding, and the C-terminal membrane fusion domain S2, which is responsible for membrane fusion [12]. During severe acute respiratory syndrome coronavirus 2 (SARS-CoV-2) entry into host cells, S protein binding to ACE2 enables its cleavage by membrane-bound TMPRSS serine proteases and subsequent fusion of the viral membrane to the host cell membrane [13].

Interferon (IFN), especially type I IFN, acts as the first line of host defense against viral infection. The induction of IFN leads to the upregulation of hundreds of IFN-stimulated genes (ISGs), which include cholesterol 25-hydroxylase (CH25H) [14]. CH25H is a 32 kDa multi-transmembrane and endoplasmic reticulum (ER)-associated enzyme that catalyzes the oxidation of cholesterol to produce 25-hydroxycholesterol (25HC) to reduce the accumulation of cholesterol [15]. 25HC is a soluble factor that controls sterol biosynthesis through the regulation of sterol-responsive element binding protein (SREBP) and nuclear receptors [16,17]. CH25H is known to play multiple biological roles, especially in lipid metabolism, inflammatory response, antiviral processes, and cell survival [18]. Recent reports revealed that CH25H inhibits various viruses, including human immune deficiency virus (HIV), hepatitis C virus (HCV), reovirus, herpes simplex virus, Zika virus (ZIV), and mouse hepatitis virus-68 [19,20,21]. These findings demonstrated that CH25H has broad-spectrum antiviral effects. The antiviral activity of CH25H is mediated by 25HC [22] and involves multiple mechanisms, including inhibiting viral replication [20,23], blocking fusion of the viral envelope with host membranes [20], and inhibiting formation of viral replication complexes on intracellular membranes [24]. 25HC is also an important factor in the regulation of lipid metabolism. Many viruses require lipid rafts, and lipid biosynthesis is important for viral replication, maturation, and secretion [25]. However, the interaction between SADS-CoV infection and CH25H or 25HC remains unknown.

The present study found that CH25H can be upregulated by SADS-CoV infection in vitro and in vivo. In addition, IFN-α induced the expression of CH25H, indicating that CH25H was an IFN-stimulated gene (ISG) in porcine ileum epithelial cells. Further, ectopic CH25H and CH25H mutants can inhibit SADS-CoV replication. We also demonstrated that 25HC significantly suppressed SADS-CoV infection via inhibiting virus entry. Notably, we finally found that CH25H and 25HC restricted SADS-CoV infection by blocking spike protein-mediated membrane fusion. These discoveries are helpful for the development of novel antiviral therapies against SADS-CoV.

## 2. Materials and Methods

### 2.1. Cells and Viruses

Porcine ileum epithelial (IPI-2I) cells, African green monkey kidney (Vero E6) cells, and human embryonic kidney (HEK293T) cells were cultured in Dulbecco’s minimal essential medium (DMEM; Invitrogen) with 10% fetal bovine serum (Invitrogen, Waltham, MA, USA) and antibiotic–antimycotic solutions (100×; Invitrogen). The cells were maintained at 37 °C in a humidified 5% CO_2_ incubator. SADS-CoV was maintained in our laboratory. SADS-CoV infected IPI-2I and Vero E6 cells as described previously [26].

### 2.2. Plasmids, Reagents, and Antibodies

DNA sequences encoding porcine CH25H (GenBank accession no. XM_021073088.1) or human CH25H (GenBank accession no. KJ897945.1) were cloned into a pCAGGS-HA vector. Porcine CH25H-M and human CH25H-M were engineered to lack catalytic activity by site-directed mutagenesis of histidine residues 242 and 243 to glutamine, and cloned into the pCAGGS-HA vector. DNA encoding codon-optimized SADS-CoV S (GenBank accession no. MT199596.1) were cloned into a pCAGGS-Myc vector. All of the plasmids described above, including pEGFP-C1, were deposited in our laboratory. 25HC (catalog no. H1015) was obtained from Sigma–Aldrich (St. Louis, MO, USA). X-tremeGENE HP DNA transfection reagent (catalog no. 6366546001) was purchased from Roche (Switzerland). Alexa Fluor 488 goat anti-mouse IgG (H + L) and Alexa Fluor 594 goat anti-rabbit IgG (H + L) secondary antibodies were obtained from Invitrogen. The SADS-CoV N protein-specific monoclonal antibody (mAb) was prepared by our laboratory [27]. A polyclonal antibody against CH25H (catalog no. PA5-70691) was purchased from Invitrogen. Rabbit anti-HA (ab9110) was purchased from Abcam (Cambridge, UK). Rabbit anti-GAPDH (G9545) was obtained from Sigma–Aldrich.

### 2.3. Western Blotting

Cells were lysed with RIPA lysis buffer (R0278; Sigma–Aldrich) and then centrifuged at 12,000 rpm for 5 min to remove cell debris. Equal amounts of total proteins were analyzed with 12.5% SDS-PAGE and transferred to nitrocellulose membranes (66485; Pall). After blocking with 5% non-fat milk for 1 h at room temperature, the nitrocellulose membranes were incubated with primary antibodies for 6–8 h at 4 °C, followed by incubation with IRDye 800CW goat anti-mouse lgG (H + L) (1:10,000) (926-32210; LiCor BioSciences, Lincoln, NE, USA) or IRDye 680RD goat anti-rabbit lgG (H + L) (1:10,000) (926-68071; LiCor BioSciences) for 45 min in the dark. An Odyssey infrared imaging system (LiCor BioSciences) was used to visualize the blots.

### 2.4. Total RNA Extraction and Quantitative Real-Time PCR (qRT-PCR)

Total RNA was extracted from treated cells using the Simply P Total RNA Extraction Kit (BioFlux, Beijing, China) and reverse transcribed into cDNA using the PrimeScript™ IV 1^st^ strand cDNA Synthesis Mix (TaKaRa, Dalian, China). Quantitative real-time PCR (qRT-PCR) was conducted using the Applied Biosystems^®^ QuantStudio^®^ 5 Real-Time PCR System (Thermo Fisher, Waltham, MA, USA) with TB green^®^ Premix Ex Taq™ II (Tli RnaseH Plus) (TaKaRa, Kusatsu-shi, Japan). The sequences of the primers synthesized for qRT-PCR are listed in Table 1. Absolute quantitative mRNA levels were calculated using standard curves, and relative quantitative mRNA levels were analyzed using the ΔΔCt method.

### 2.5. Animal Experiment and Immunohistochemistry (IHC) Assay

Six three-day-old specific pathogen-free (SPF) piglets were randomly divided into two groups with three piglets in each group. The SPF piglets in group 1 were orally challenged with the SADS-CoV strain of 5 × 10^4^ TCID_50_. The SPF piglets in group 2 were orally challenged with DMEM, serving as uninfected controls. After inoculation, the piglets were observed and recorded thrice daily for clinical symptoms of vomiting, diarrhea, lethargy, and body condition. All piglets were euthanized according to the Ethical Committee of the Institute, which was terminated at 48 h post-infection (hpi). Representative sections of the ileal tissues were fixed with 4% paraformaldehyde and stored in 70% ethanol at 4 °C. The IHC assay was performed as previously described [28]. Slides were incubated with mAb 3E9 (1:50) at 4 °C overnight and subsequently incubated with HRP-labelled goat anti-mouse IgG (Sigma-Aldrich, AP308P) for 1 h. Immunocomplexes were detected using the 3,3′-diaminobenzidine liquid substrate system.

### 2.6. Median Tissue Culture Infectious Dose Assay (TCID_50_) and Cell Viability Assay

Vero E6 cells were cultured in 96-well plates to 90% confluency and infected with 10-fold graded dilutions of the virus. The medium was removed after the cells were incubated with viruses for 1 h at 37 °C. The cells were incubated with DMEM containing 5 μg/mL trypsin. At 3–5 days post-infection, the cytopathic effect was observed under an inverted microscope, and the virus titers were calculated using the Reed–Muench method. IPI-2I and Vero E6 cells cultured in 96-well plates were treated with 25HC at different concentrations (1, 2.5, 5, 10, and 20 μM) or absolute ethanol control for 24 h. Cell viability was detected using Cell Counting Kit-8 (CCK-8) (CK04; Dojindo, Rockville, MD, USA) according to the manufacturer’s instructions.

### 2.7. Small Interfering RNA Assay

Small interfering RNAs (siRNAs) targeting the porcine CH25H gene were designed and synthesized by Guangzhou Ribobio Co., LTD (Guangzhou, China). The siRNA target sequences of CH25H were as follows: siRNA1 (5′-CTACATCACTCCCAGTTTA-3′), siRNA2 (5′-TCCTGATCTTCCACGTGAT-3′), and siRNA3 (5′-ACAAAGTGCCTTGGCTGTA-3′). IPI-2I cells were seeded in a 12-well plate and grown to 40–50% confluence. The siRNAs and negative control siRNA (NC) were transfected into the cells using Lipofectamine RNAiMAX transfection reagent (13778150; Invitrogen). At 48 h post-transfection (hpt), the cells were infected with SADS-CoV (multiplicity of infection (MOI) 0.1], and cell lysates and supernatants were harvested for protein, mRNA, and virus titration analyses.

### 2.8. Immunofluorescence Assay (IFA)

IPI-2I and Vero E6 cells were fixed with 4% paraformaldehyde for 30 min at room temperature. After washing with PBS three times, the cells were permeabilized with 0.25% Triton X-100 in PBS for 15 min and blocked with 5% non-fat milk in PBS for 1 h at room temperature. After three washes with PBS, the cells were incubated with mouse anti-SADS-CoV N mAb (1:100 dilution) at 4 °C overnight and then with goat anti-mouse secondary antibody (1:300 dilution) conjugated to Alexa Fluor 488 (Invitrogen) for 45 min in the dark. After washing with PBS three times, nuclei were stained with 4′, 6-diamidino-2-phenylindole for 15 min in the dark. The fluorescent images were obtained with an inverted fluorescence microscope (EVOS M5000, Life, USA). Quantification of the fluorescence-positive cells was performed by taking the average of at least six fields of view.

### 2.9. Membrane Fusion Assay

Vero E6 cells or HEK293T cells were seeded in a 12-well plate. Cells were transfected with pEGFP-C1 and Myc-SADS-CoV S-expressing plasmids using X-tremeGENE HP DNA transfection reagent (6366546001; Roche, Switzerland). Where indicated, plasmids encoding human CH25H were co-transfected. In addition, cells were pretreated with 25HC for 1h, followed by co-transfected plasmids expressing EGFP and SADS-CoV S. For HEK293T cells, the cells were fixed and stained with DAPI at 24 hpt; for Vero E6 cells, the cells were fixed and stained with DAPI at 36 hpt. Cells were then observed and photographed with an inverted fluorescence microscope (EVOS M5000, Life, USA). Quantification of membrane fusion induced by SADS-CoV S protein was performed by calculating the number of cells in GFP+ syncytia.

### 2.10. Statistical Analysis

All results shown in the figures were presented, where appropriate, as mean and standard deviation (SD). The results of three independent experiments were analyzed with GraphPad Prism 8 (GraphPad Software). *P* values were calculated using the two-tailed unpaired Student’s *t*-test. Differences were considered significant if *p* was <0.05.

## 3. Results

### 3.1. SADS-CoV Infection Induces CH25H Expression In Vitro and In Vivo

To examine CH25H expression, IPI-2I and Vero E6 cells were infected with SADS-CoV (MOI 0.25, 0.5, or 1) for 24 h. Protein and mRNA levels of CH25H were analyzed by Western blotting and qRT-PCR. Protein (Figure 1A) or mRNA (Figure 1B) levels of CH25H in SADS-CoV-infected cells exhibited a significant increase compared with the control group. We infected IPI-2I and Vero E6 cells with SADS-CoV (MOI 1) and collected cell samples at 6, 12, and 24 hpi so that CH25H expression could be measured. Compared with uninfected control cells, SADS-CoV infection promoted CH25H protein (Figure 1C) and mRNA (Figure 1D) expression at all time points. Next, we explored whether SADS-CoV infection induces CH25H expression in vivo. Three-day-old SPF piglets were orally challenged with SADS-CoV, and unchallenged piglets served as mock controls. At 48 hpi, ileal tissues from challenged and unchallenged piglets were harvested. Successful infection of ileal tissue by SADS-CoV was confirmed by IHC (Figure 1E). Total RNA was extracted from ileal tissues, and CH25H mRNA levels were analyzed by qRT-PCR. As anticipated, the CH25H level was significantly increased in the ileum of SADS-CoV-infected piglets compared to uninfected piglets (Figure 1F). These analyses revealed that SADS-CoV infection upregulates CH25H expression in vitro and in vivo.

### 3.2. CH25H Is an IFN-Stimulated Gene in IPI-2I Cells

Previous studies reported that CH25H was an ISG in multiple cell types, such as in chicken peripheral blood mononuclear cells, embryo fibroblast cell lines (DF1), mice bone marrow-derived macrophages (BMDMs), and dendritic cells [29]. However, it was not an ISG in Vero cells [30]. To illustrate whether CH25H was an ISG in IPI-2I cells, IFN-α was used to induce mRNA expression of intracellular CH25H and ISG15 genes. IPI-2I cells were treated with IFN-α (100 and 1000 μg/mL) for 12 h. The cell samples were collected to detect the mRNA levels using qRT-PCR. The mRNA levels of CH25H and ISG15 in IPI-2I cells treated with IFN-α were increased compared with untreated cells (Figure 1A,B). Then, IPI-2I cells were treated with 1000 μg/mL IFN-α and incubated for 0, 2, 4, 6, or 12 h. CH25H mRNA level first increased and then decreased after IFN-α treatment, and reached the highest level at about 6 h (Figure 2C). The mRNA level of ISG15 (Figure 2D) was also increased. In conclusion, the expression of CH25H was induced by IFN-α, indicating that CH25H was an ISG in IPI-2I cells.

### 3.3. Knockdown of CH25H Facilitates SADS-CoV Replication

To investigate whether CH25H was a host restriction factor in SADS-CoV-infected IPI-2I cells, three siRNAs were used to downregulate endogenous CH25H. The siRNAs were transfected into IPI-2I cells, either alone or in combination, and the NC was used as a control. CH25H protein and mRNA levels were examined by Western blotting and qRT-PCR at 48 hpt. The data demonstrate that all three siRNAs were able to restrict the expression of CH25H, and the most effective inhibition occurred in the three siRNAs co-treatment group (Figure 3A,B). IPI-2I cells were transfected with the siRNAs for 48 h, followed by infection with SADS-CoV (MOI 0.1). After 24 h, cells were harvested for Western blotting and qRT-PCR, and supernatants were examined by TCID_50_ assay. Our results revealed that the levels of SADS-CoV N protein (Figure 3C) and mRNA (Figure 3D) were increased in response to the interference of CH25H, compared with the NC group. TCID_50_ further manifested that the downregulation of CH25H significantly increased viral titers (Figure 3E). These results suggest that CH25H served as a host restriction factor for SADS-CoV infection and that suppression of endogenous CH25H was beneficial to viral replication.

### 3.4. Overexpression of CH25H Inhibits Replication of SADS-CoV

To elucidate the effect of CH25H on the replication of SADS-CoV, plasmids encoding porcine CH25H (pCH25H) with an HA tag were constructed. IPI-2I cells were transfected with pCH25H plasmids or an empty vector, followed by infection with SADS-CoV (MOI 0.1) for 24 h. The expression of SADS-CoV N protein was decreased in pCH25H-treated groups compared with the control (Figure 4A). In addition, a marked reduction in SADS-CoV N mRNA levels (Figure 4B) and viral titers (Figure 4C) was observed. We next constructed a plasmid encoding human CH25H (hCH25H) with an HA tag and extended our findings to Vero E6 cells. Not surprisingly, we found that SADS-CoV N protein expression (Figure 4D), mRNA level (Figure 4E), and virus titer (Figure 4F) were markedly decreased. Thus, our findings demonstrate that overexpression of CH25H significantly suppressed the replication of SADS-CoV in a dose-dependent manner.

### 3.5. CH25H-M Lacking Hydroxylase Activity Can Inhibit SADS-CoV Replication

As previously reported, histidine residues 242 and 243 of CH25H are critical for its enzymatic activity [31,32]. To corroborate whether the enzyme activity of CH25H was indispensable for its activity against SADS-CoV, plasmids of porcine CH25H mutant (pCH25H-M) and human CH25H mutant (hCH25H-M) with an HA tag were generated according to converting histidine residues 242 and 243 to glutamine (H242Q and H243Q). Plasmids expressing pCH25H or pCH25H-M proteins were transfected into IPI-2I cells. At 24 hpt, the cells were infected with SADS-CoV (MOI 0.1), and cell samples and supernatants were harvested at 24 hpi. Western blotting (Figure 5A) and qRT-PCR (Figure 5B) showed that both pCH25H or pCH25H-M downregulated expression of SADS-CoV N protein and mRNA in contrast to the Mock-transfected group, and a significant inhibitory effect of CH25H, compared with CH25H-M, was observed after SADS-CoV infection. Quantification of the virus by TCID_50_ (Figure 5C) demonstrated that the viral titer of CH25H was lower than that of CH25H-M in IPI-2I cells. Similar results were observed in Vero E6 cells. The levels of SADS-CoV N protein (Figure 5D) mRNA (Figure 5E) and viral titers (Figure 5F) in hCH25H- and hCH25H-M-transfected cells were decreased compared with untreated cells. These results demonstrate that CH25H-M lacking hydroxylase activity still inhibited SADS-CoV replication, although not as potently as CH25H.

### 3.6. 25HC Inhibits SADS-CoV Infection

The CH25H gene encodes an enzyme which catalyzes the formation of 25HC from cholesterol [32]. 25HC has been shown to have broad antiviral activity by inhibiting the host cell entry of porcine reproduction and respiratory syndrome virus (PRRSV), ZIV, PEDV, PDCoV, SARS-CoV, SARS-CoV-2, and Middle East respiratory syndrome coronavirus (MERS-CoV). [22,30,33,34]. Therefore, to verify whether 25HC inhibits SADS-CoV infection, we treated IPI-2I and Vero E6 cells with 25HC (1, 2.5, 5, 10, and 20 μM) as well as ethanol as a control, and detected cell viability by CCK-8 assay. Cell viability was not significantly changed by treatment with 10 μM 25HC in both cell lines (Figure 6A), which provided the foundation for further research into the relationship between 25HC and SADS-CoV infection. IPI-2I and Vero E6 cells were pretreated with 25HC (5 and 10 μM) for 1 h, or with ethanol as a control. We infected cells with SADS-CoV (MOI 0.1) and observed the levels of viral N protein and mRNA at 24 hpi. 25HC treatment dramatically reduced the expression of viral N protein (Figure 6B) and mRNA (Figure 6C) compared with the controls. The viral titers (Figure 6D) in the supernatants from SADS-CoV-infected IPI-2I and Vero E6 cells were decreased when the 25HC concentration was increased. Similar to the above studies, the IFA (Figure 6E,F) showed that the amount of red fluorescence from the 25HC-treated cells was less than in the mock cells, which suggested that the 25HC treatment suppressed viral replication in both cell types. As anticipated, our results revealed that 25HC inhibited SADS-CoV infection in a dose-dependent manner.

### 3.7. 25HC Restricts SADS-CoV Infection by Blocking Viral Entry

Based on the above findings, elucidating the antiviral mechanism of 25HC against SADS-CoV was essential for subsequent research. Therefore, we investigated from two aspects: (1) to detect whether 25HC treatment directly inactivated SADS-CoV; and (2) which phase, during the SADS-CoV life cycle of attachment, internalization, replication, and release, was blocked by 25HC. First, we investigated the activity of virions by treating them directly with 25HC. SADS-CoV (MOI 0.1) was mixed with 25HC (10 μM) or ethanol containing 5 μg/mL trypsin for 3 h at 37 °C. IPI-2I and Vero E6 cells were prechilled for 1 h at 4 °C and then replaced by a mixture of 25HC or ethanol and SADS-CoV at 4 °C for another 2 h. After being washed three times with PBS, DMEM containing 5 μg/mL trypsin was added to the media, and the mixture was incubated for 36 h at 37 °C. Cells were collected to examine the SADS-CoV N mRNA levels by qRT-PCR. No obvious alteration of SADS-CoV N mRNA level was observed in 25HC-treated cells compared with other untreated cells (Figure 7A,F).

Subsequently, the influence of 25HC on SADS-CoV attachment was analyzed. IPI-2I and Vero E6 cells were pretreated with 25HC (10 μM) or ethanol for 1 h at 37 °C. Cells were then treated with the mixture of 25HC (10 μM) or ethanol and SADS-CoV (MOI 0.1) containing 5 μg/mL trypsin and incubated for 1 h at 4 °C. The mRNA level of the SADS-CoV N was determined from the cells. Viral mRNA levels in 25HC-treated cells did not differ from those in other untreated cells, suggesting that the attachment phase of SADS-CoV was not affected by 25HC (Figure 7B,G). We investigated the internalization stage of SADS-CoV. IPI-2I and Vero E6 cells were prechilled for 1 h at 4 °C, followed by infection with SADS-CoV (MOI 0.1) in the presence of 5 μg/mL trypsin. Both cell types were incubated for 1 h at 4 °C. After removal of unbound virions by extensive washing with chilled DMEM, the culture medium was replaced with DMEM containing 25HC (10 μM) or ethanol and inoculated at 37 °C for another 2 h. The cells were washed with DMEM and collected to determine SADS-CoV N mRNA levels. A significant reduction in SADS-CoV N mRNA level was observed in 25HC-treated cells but not in other untreated cells (Figure 7C,H). The results indicate that 25HC suppressed SADS-CoV infection by blocking viral internalization.

To evaluate whether 25HC affected the replication of SADS-CoV, both cell types were infected with SADS-CoV at MOI 0.1 at 37 °C. After 6 hpi, the culture medium was replaced with DMEM, and the cells were treated with 25HC (10 μM) or ethanol for 10 h at 37 °C. The levels of SADS-CoV N mRNA obtained from the cell supernatants were then measured. 25HC treatment showed no distinct change in SADS-CoV N mRNA level compared with other untreated cells (Figure 7D,I), indicating that 25HC failed to perturb the replication of SADS-CoV. Finally, we investigated the effect of 25HC on viral release. Both cell types were infected with SADS-CoV (MOI 0.1) for 16 h, and the cell supernatants were collected after treatment with 25HC (10 μM) or ethanol for 1 h. Similar to the above findings, the levels of SADS-CoV N mRNA between cells in the presence or absence of 25HC did not differ (Figure 7D,J), illustrating that the release of SADS-CoV was not influenced by 25HC. These results indicate that 25HC restricted SADS-CoV infection by inhibiting viral entry.

### 3.8. CH25H and 25HC Block S Protein-Mediated Membrane Fusion

CH25H and 25HC exert an antiviral effect against numerous viruses through multiple mechanisms [29]. For example, 25HC inhibits HCV replication by preventing the formation of a membranous web [24] and restrains SARS-CoV-2 reproduction by disturbing membrane fusion [13]. We next examined the effect of CH25H and 25HC on SADS-CoV S protein-mediated membrane fusion, as coronavirus entry is largely controlled by the S protein with receptor binding and membrane fusion capabilities [10]. We first analyzed the expression of SADS-CoV S. Vero E6 cells were transfected with Myc-SADS-CoV S-expressing plasmids, accompanied by an empty vector group. The expression of SADS-CoV S protein was assessed by Western blotting 36 hpt (Figure 8A). Moreover, independent of virus infection, an in vitro cell membrane fusion assay based on the expression of SADS-CoV S, EGFP, and hCH25H was established. Meanwhile, we also co-transfected SADS-CoV S and EGFP plasmids in the presence or absence of 25HC (10 μM) into Vero E6 and HEK293T cells, accompanied by control groups. For Vero E6 cells, at 36 hpt, the cells were fixed and stained with DAPI. Cell membrane fusion mediated by SADS-CoV S is substantially reduced due to CH25H expression or 25HC treatment (Figure 8B). Compared with 25HC treatment, the inhibitory effect of CH25H on S protein-mediated membrane fusion was weaker (Figure 8C). In addition, the same result was observed in HEK293T cells (Figure 8B,C). These data manifest that CH25H and 25HC blocked S protein-mediated membrane fusion.

## 4. Discussion

SADS-CoV is a newly discovered porcine enteric coronavirus in southern China, which can cause severe and acute diarrhea and rapid weight loss in piglets [3]. The pathogenic mechanisms of SADS-CoV infection have so far not been fully characterized [35]. As a molecular mediator of innate antiviral immunity, CH25H encodes 272 and 270 amino acids in human and porcine cells and converts cholesterol to 25HC [36]. Recent evidence suggests that CH25H and 25HC also act as important regulators of inflammation, immunity, and antiviral activity [29]. CH25H and 25HC have an antiviral effect against enveloped and non-enveloped viruses [29,37]. However, the interactions between SADS-CoV infection and CH25H or 25HC have not been reported. In this study, we were the first to identify the function and mechanism of CH25H and its catalytic product 25HC in suppressing SADS-CoV infection.

Previous studies have shown that CH25H can be induced by HCV, PRRSV, PDCoV, and SARS-CoV-2 [22,24,33,34,38]. In contrast, several studies found that CH25H can be downregulated by herpes simplex virus-1 and PEDV [21,30]. Here, the in vitro and in vivo experiments demonstrated that SADS-CoV infection resulted in a significant increase in CH25H expression. This suggests that CH25H plays a crucial role during SADS-CoV infection. IFNs are potent activators of antiviral factors, which induce the expression of ISGs that have a variety of functions ranging from direct inhibition of viral components to activation of other immune cell types [14]. IFN-α (type I) and IFN-γ (type II) induce the expression of murine CH25H in BMDMs [20]. A recent study suggested that porcine CH25H was induced by type I IFN (IFN-α and IFN-β) and that IFN-γ promoted CH25H expression in porcine macrophage 3D4/21 cells [38]. In the present study, the expression of CH25H was induced by IFN-α in IPI-2I cells, which indicated that CH25H is an ISG. Knockdown of CH25H contributed to SADS-CoV infection in IPI-2I cells, suggesting that CH25H is a host restriction factor. Unfortunately, no prominent knockdown of CH25H was observed in Vero E6 cells because the sequence of siRNAs targeting porcine CH25H failed to target human CH25H.

Here, we found that overexpression of pCH25H and hCH25H restrained SADS-CoV infection in IPI-2I and Vero E6 cells. To investigate whether the enzyme activity of CH25H was indispensable for its activity against SADS-CoV, we next examined the antiviral activity of CH25H-M, a mutant lacking hydroxylase activity. Similar results were observed in both cell lines: CH25H-M attenuated the proliferation of SADS-CoV, but the inhibitory activity was significantly reduced compared with that of CH25H. Our study confirmed for the first time that infection of SADS-CoV was inhibited by CH25H; however, this suppression did not completely rely on the enzymatic activity of CH25H. Previous studies have reported that CH25H affects the proliferation of different viruses through different mechanisms in a 25HC-dependent or 25HC-independent manner. For example, CH25H inhibits pseudorabies virus by blocking viral attachment and internalization [38]; CH25H/CH25H-M reduces PRRSV replication by degrading the nonstructural protein 1 alpha of PRRSV through the ubiquitin-proteasome pathway [39]. 

The previous study reported that CH25H inhibits HCV replication via 25HC-mediated disruption of the function of SREBP2, further acting on the late stage of the HCV replication cycle, eventually effectively inhibiting virus infection [40]. Chen et al. suggested that the antiviral effect of CH25H against HCV does not depend entirely on its enzyme activity, as a mutant form of CH25H still reduced HCV infection by interacting with and suppressing NS5A dimerization [25]. In the present study, the overexpression of CH25H inhibited SADS-CoV infection, which is similar to that observed in PRRSV. In addition, the catalytically inactive form of CH25H retains antiviral action against PRRSV and HCV [25]. Therefore, our data further demonstrate that CH25H served as a host restriction factor for SADS-CoV infection. As the catalytic product of CH25H, 25HC has been reported to exert a broad range of antiviral activity [41], including against porcine enteric coronaviruses (PEDV, TGEV, and PDCoV) [30,34] and human coronaviruses (SARS-CoV, SARS-CoV-2, and MERS-CoV) [22]. Consistent with previous studies, we found that replication of SADS-CoV could be restrained by 25HC, which provided the basis for our investigation of the antiviral mechanism of CH25H. The results suggest that CH25H resists SADS-CoV infection in a 25HC-dependent and 25HC-independent manner. 25HC exerts an antiviral effect against numerous viruses through multiple mechanisms [29]. For example, 25HC suppressed ZIKV, PRRSV, PEDV, and PDCoV by blocking viral entry [30,33,34,42]. 25HC inhibited HCV replication, preventing the formation of a membranous web [24], and restrained SARS-CoV-2 reproduction by disturbing membrane fusion [13]. Wang indicated that 25HC through depleting membrane cholesterol suppressed human coronaviruses (SARS-CoV, SARS-CoV-2, and MERS-CoV) [22]. Coronavirus S protein mediates entry into cells by recognizing cell receptors and catalyzing fusion between the viral envelope and the cell membrane [10]. As a member of the coronavirus family, SADS-CoV is similar to other coronaviruses in S protein-mediated membrane fusion. Here, we show that CH25H and 25HC blocked S protein-mediated membrane fusion in Vero E6 and HEK293T cells, and that the inhibitory fusion effect of CH25H was weaker than that of 25HC treatment.

## 5. Conclusions

Taken together, our data demonstrate that SADS-CoV was inhibited by CH25H and CH25H-M. Moreover, we indicated that 25HC significantly suppressed SADS-CoV replication. Further investigation found that CH25H and 25HC restrained SADS-CoV proliferation by blocking S protein-mediated membrane fusion. These findings are helpful for the development of novel therapies against SADS-CoV.

## Figures and Tables

**Figure 1 viruses-15-02406-f001:**
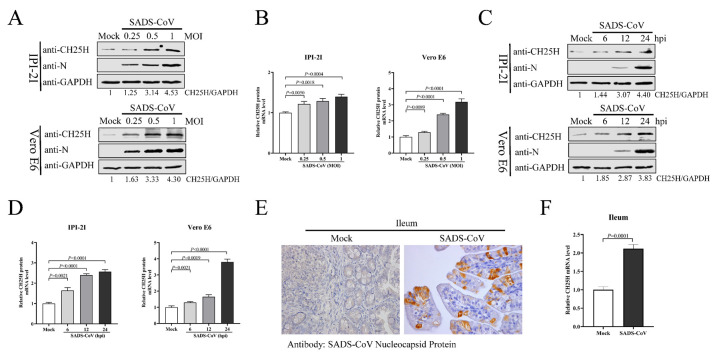
SADS-CoV infection induced CH25H expression in vitro and in vivo. (**A**,**B**) IPI-2I and Vero E6 cells were infected with different doses of SADS-CoV (MOI 0.25, 0.5, or 1). Uninfected cells were used as a control group. Samples from both cell types were harvested at 24 h, and CH25H protein and mRNA levels were determined by Western blotting (**A**) and qRT-PCR (**B**), respectively. (**C**,**D**) IPI-2I and Vero E6 cells were infected with SADS-CoV at MOI 1. Cell samples were collected at 6, 12, and 24 hpi. CH25H protein and mRNA levels were detected by Western blotting (**C**) and qRT-PCR (**D**). (**E**) Representative microphotographs of viral antigen immunochemical staining in SADS-CoV-uninfected and -infected ileal tissues (Bar: 50 μm). (**F**) Total RNA was extracted from ileal tissues, and CH25H mRNA levels were analyzed by qRT-PCR. Means and SD (error bars) of three independent experiments are indicated. *p* values were calculated using two-tailed unpaired Student’s *t*-test.

**Figure 2 viruses-15-02406-f002:**
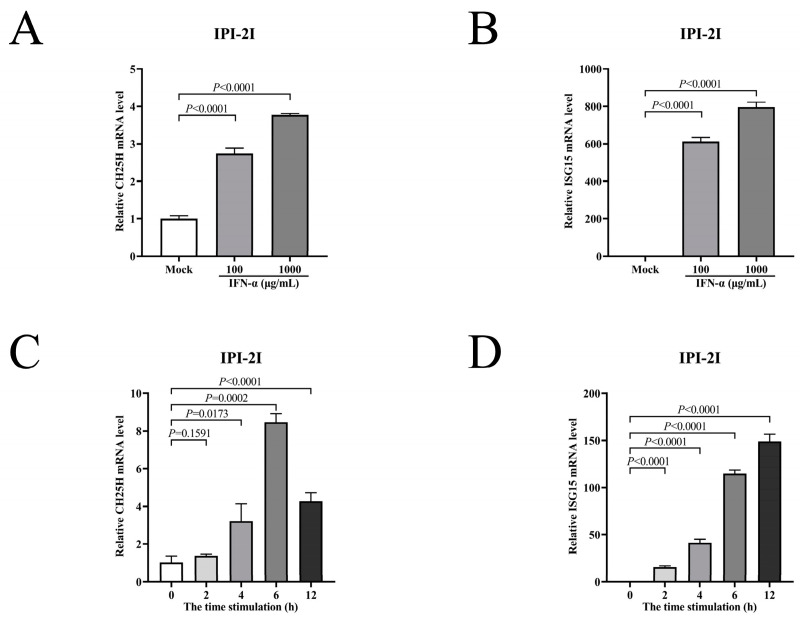
CH25H is an IFN-stimulated gene in IPI-2I cells. (**A**,**B**) IPI-2I cells were treated with IFN-α (100 and 1000 μg/mL) for 12 h, accompanied by a control group. Cell samples were harvested to detect the mRNA levels of CH25H (**A**) and ISG15 (**B**) using qRT-PCR. (**C**,**D**) IPI-2I cells were treated with 1000 μg/mL IFN-α, and cell samples were collected at 0, 2, 4, 6, and 12 h. mRNA levels of CH25H (**C**) and ISG15 (**D**) were analyzed by qRT-PCR. Means and SD (error bars) of three independent experiments are indicated. *p* values were calculated using two-tailed unpaired Student’s *t*-test.

**Figure 3 viruses-15-02406-f003:**
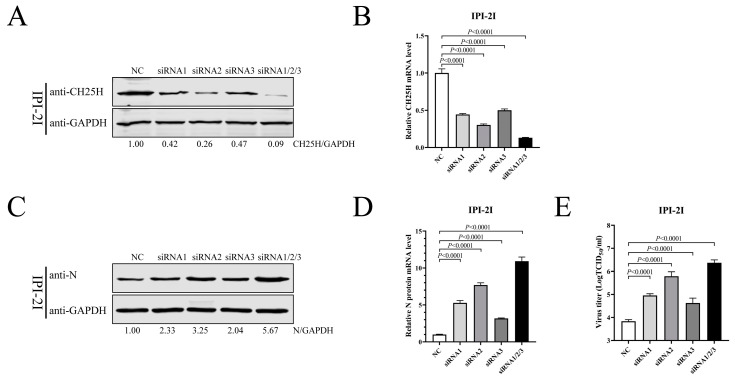
Knockdown of CH25H facilitates SADS-CoV replication. (**A**,**B**) IPI-2I cells were transfected with siRNAs and NC. Cell samples were harvested at 48 hpt. CH25H protein (**A**) and mRNA (**B**) were then examined by Western blotting and qRT-PCR. (**C**–**E**) IPI-2I cells were transfected with siRNAs for 48 h, followed by infection with SADS-CoV (MOI 0.1). At 24 hpi, cell samples were collected to determine SADS-CoV N protein expression via Western blotting (**C**) and qRT-PCR (**D**), and viral titers were detected by TCID_50_ assay from supernatant (**E**). Means and SD (error bars) of three independent experiments are indicated. *p* values were calculated using two-tailed unpaired Student’s *t*-test.

**Figure 4 viruses-15-02406-f004:**
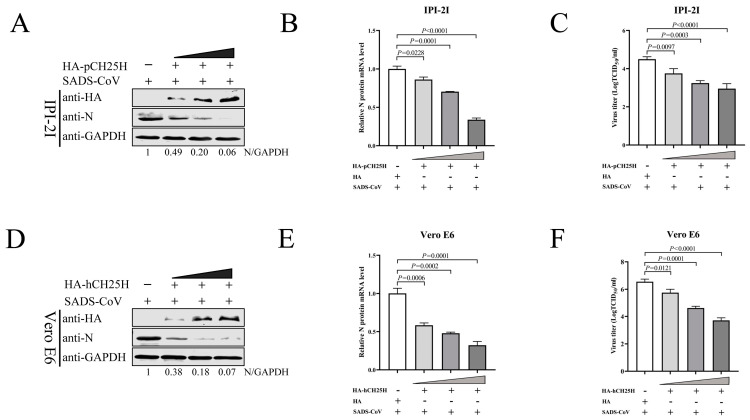
Overexpression of CH25H inhibited replication of SADS-CoV. (**A**–**C**) IPI-2I cells were transfected with porcine CH25H plasmids (HA-pCH25H) at 0.5, 1, and 2 μg, accompanied by an empty vector group. Cells were infected with SADS-CoV at MOI 0.1 at 24 h post-transfection. After 24 hpi, the cell samples were collected to detect CH25H and SADS-CoV N protein expression by western blotting (**A**). qRT-PCR analyzed the mRNA levels of SADS-CoV N (**B**). SADS-CoV titers were examined by TCID_50_ assay from supernatants (**C**). (**D**–**F**) Vero E6 cells were transfected with human CH25H plasmids (HA-hCH25H) at 0.5, 1, and 2 μg, accompanied by an empty vector group. Cells were infected with SADS-CoV at MOI 0.1 at 24 h post-transfection. After 24 hpi, the cell samples were collected to detect CH25H and SADS-CoV N protein expression by western blotting (**D**). qRT-PCR analyzed the mRNA levels of SADS-CoV N (**E**). SADS-CoV titers were examined by TCID_50_ assay from supernatants (**F**). Means and SD (error bars) of three independent experiments are indicated. *p* values were calculated using two-tailed unpaired Student’s *t*-test.

**Figure 5 viruses-15-02406-f005:**
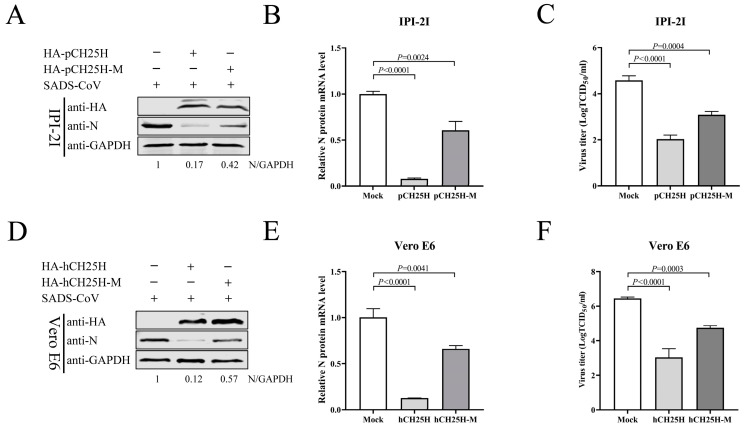
CH25H-M lacking hydroxylase activity can inhibit SADS-CoV replication. (**A**–**C**) IPI-2I cells were transfected with 3 μg HA-pCH25H, HA-pCH25H-M, and an empty vector for 24 h, followed by infection with SADS-CoV (MOI 0.1) for 24 h. Samples were harvested to examine the protein expression of CH25H and SADS-CoV N by Western blotting (**A**). qRT-PCR detected the mRNA levels of SADS-CoV N (**B**). SADS-CoV titers were determined by TCID_50_ assay from supernatants (**C**). (**D**–**F**) Vero E6 cells were transfected with the plasmids expressing HA-hCH25H, HA-hCH25H-M, and empty vector. At 24 h post-transfection, the cells were infected with SADS-CoV (MOI 0.1) at 24 hpi. Samples were harvested to examine the protein expression of CH25H and SADS-CoV N by Western blotting (**D**). qRT-PCR detected mRNA levels of SADS-CoV N (**E**). SADS-CoV titers were determined by TCID_50_ assay from supernatants (**F**). Means and SD (error bars) of three independent experiments are indicated. *p* values were calculated using two-tailed unpaired Student’s *t*-test.

**Figure 6 viruses-15-02406-f006:**
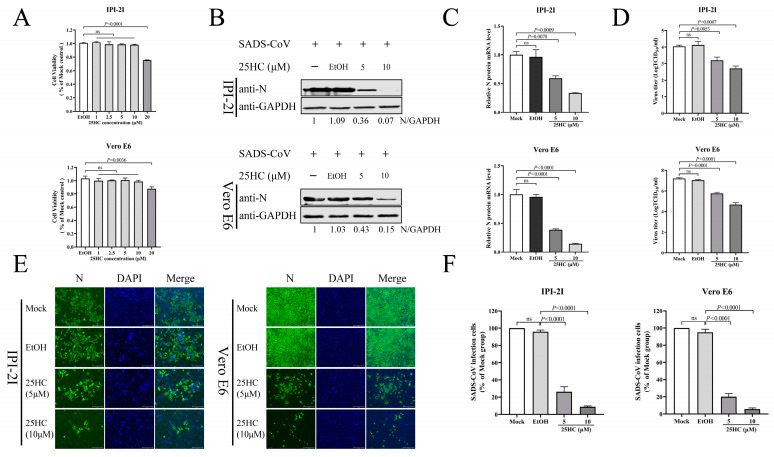
25HC inhibited SADS-CoV infection. (**A**) IPI-2I and Vero E6 cells were treated with 25HC (1, 2.5, 5, 10, or 20 μM) for 24 h. Ethanol-treated cells were used as a control group. Cytotoxicity was detected by CCK-8 assay. (**B**–**D**) Both cell types were pretreated with 25HC at 5 and 10 μM for 1 h at 37 °C. We replaced the culture medium before infection with SADS-CoV at MOI 0.1 for 24 h. We harvested the cell samples and supernatants to examine protein and mRNA levels of SADS-CoV N and viral titers using Western blotting (**B**), qRT-PCR (**C**), and TCID_50_ (**D**), respectively. (**E**) Immunofluorescence assays were used to detect SADS-CoV infection using an anti-SADS-CoV N protein antibody. The cell nuclei were stained with DAPI. Scale bar = 125 μm. (**F**) We quantified SADS-CoV-infected cells by measuring the level of red fluorescence compared with that in the mock-infected cells. Means and SD (error bars) of three independent experiments are indicated. *p* values were calculated using two-tailed unpaired Student’s *t*-test. ns means not significant.

**Figure 7 viruses-15-02406-f007:**
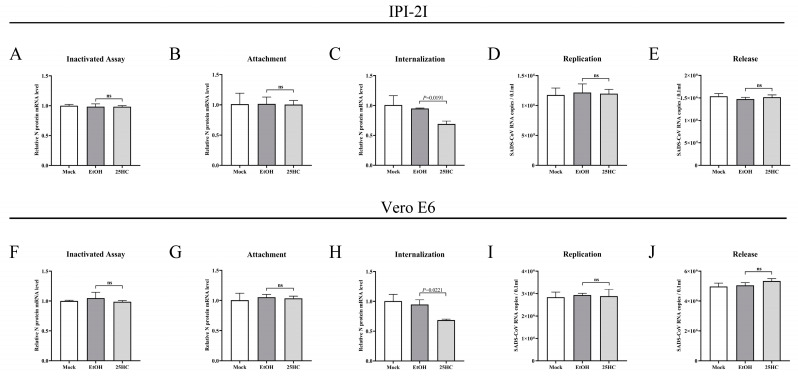
25HC suppressed SADS-CoV infection by blocking viral entry. (**A**,**F**) Inactivation assay. SADS-CoV (MOI 0.1) was mixed with 25HC (10 μM) or ethanol containing 5 μg/mL trypsin for 3 h at 37 °C. Cells were prechilled for 1 h at 4 °C and then replaced by a mixture of 25HC or ethanol and SADS-CoV at 4 °C for another 2 h. After being washed three times with PBS, DMEM containing 5 μg/mL trypsin was added to the media, and the mixture was incubated for 36 h at 37 °C. Cells were collected to examine the SADS-CoV mRNA by qRT-PCR. (**B**,**G**) Attachment assay. IPI-2I and Vero E6 cells were pretreated with 25HC (10 μM) and ethanol for 1 h at 37 °C. Cells were then treated with the mixture of 25HC (10 μM) or ethanol and SADS-CoV (MOI 0.1) containing 5 μg/mL trypsin and incubated for 1 h at 4 °C. Cells were harvested to determine SADS-CoV mRNA by qRT-PCR. (**C**,**H**) Internalization assay. IPI-2I and Vero E6 cells were prechilled for 1 h at 4 °C, followed by infection with SADS-CoV (MOI 0.1) in the presence of 5 μg/mL trypsin. Both cell types were then incubated for 1 h at 4 °C. After removal of unbound virions by extensive washing with chilled DMEM, the culture medium was replaced with DMEM in the presence of 25HC (10 μM) or ethanol and inoculated at 37 °C for 2 h. Cells were washed with DMEM three times and collected for analysis of SADS-CoV N mRNA levels by qRT-PCR. (**D**,**I**) Replication assay. IPI-2I and Vero E6 cells infected with SADS-CoV (MOI 0.1) containing 5 μg/mL trypsin, respectively. Cells were incubated for 6 h at 37 °C and washed with DMEM. Both cell types were treated with 25HC (10 μM) or ethanol for 10 h at 37 °C. Cell supernatants were harvested and SADS-CoV mRNA was measured by qRT-PCR. (**E**,**J**) Release assay. IPI-2I and Vero E6 cells were infected with SADS-CoV (MOI 0.1) containing 5 μg/mL trypsin for 16 h at 37 °C. The culture medium was replaced with DMEM in the presence of 25HC (10 μM) and ethanol and inoculated for 1 h at 37 °C. The cell supernatants were harvested and SADS-CoV mRNA was determined via qRT-PCR. Means and SD (error bars) of three independent experiments are indicated. *p* values were calculated using two-tailed unpaired Student’s *t*-test. ns means not significant.

**Figure 8 viruses-15-02406-f008:**
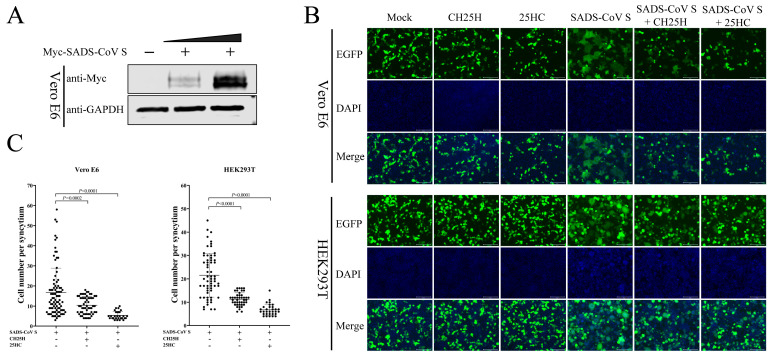
CH25H and 25HC blocked S protein-mediated membrane fusion. (**A**) Vero E6 cells were transfected with 1 and 2 μg Myc-SADS-CoV S-expressing plasmids, accompanied by an empty vector group. The expression of S protein was examined by Western blotting using an anti-Myc antibody 36 hpt. (**B**) Cells were co-transfected with 2 μg Myc-SADS-CoV S-expressing plasmids and 0.5 μg pEGFP-C1 plasmids, accompanied by control groups. Where indicated, plasmids encoding human CH25H (1 μg) were co-transfected. In addition, cells were pretreated with 25HC for 1h, followed by co-transfected plasmids expressing EGFP and SADS-CoV S. For HEK293T cells, the cells were fixed and stained with DAPI at 24 hpt; for Vero E6 cells, the cells were fixed and stained with DAPI at 36 hpt. Cells were then observed and photographed with an inverted fluorescence microscope. Scale bar = 125 μm. (**C**) Quantification of membrane fusion induced by SADS-CoV S protein was performed by calculating the number of cells in GFP+ syncytia. Means and SD (error bars) of three independent experiments are indicated. *p* values were calculated using two-tailed unpaired Student’s *t*-test.

**Table 1 viruses-15-02406-t001:** Sense sequences are used for qRT-PCR in this study.

Target	Sense	Sequences (5′-3′)
SADS-CoV N	Forward Reverse	CCCCTAAACCGGCTCGTAA CAGAATTAGGAACACGCTTCCA
hCH25H	Forward Reverse	CCCCTAAACCGGCTCGTAA CAGAATTAGGAACACGCTTCCA
pCH25H	Forward Reverse	CCCCTAAACCGGCTCGTAA CAGAATTAGGAACACGCTTCCA
hGAPDH	Forward Reverse	CCCCTAAACCGGCTCGTAA CAGAATTAGGAACACGCTTCCA
pGAPDH	Forward Reverse	CCCCTAAACCGGCTCGTAA CAGAATTAGGAACACGCTTCCA
pISG15	Forward Reverse	CCCCTAAACCGGCTCGTAA CAGAATTAGGAACACGCTTCCA

## Data Availability

The data presented in this study are available on request from the corresponding authors.

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
