# Peer review of "Cholesterol 25-Hydroxylase Suppresses Swine Acute Diarrhea Syndrome Coronavirus Infection by Blocking Spike Protein-Mediated Membrane Fusion"

_viruses, 2023, doi:10.3390/v15122406_

Round 1
Reviewer 1 Report
Comments and Suggestions for Authors
In this manuscript, Lui D. et al provided a comprehensive series of experiments to define the role of CH25H in SADS-CoV replication/entry largely using the cellular model and a few in vivo experiments. Authors showed that CH25H is an ISG, its expression induced in response to viral infection. CH25H is a cholesterol hydrolysing enzyme which converts cholesterol into 25HC. Both CH25H and 25-HC inhibit the virus replication and spiked-mediated syncytia formation in dose dose-dependent manner. Overall, the present manuscript is well conceived, planned and executed. However, there are a few concerns which must be addressed to improve the quality of the manuscript.
1. 25HC inhibits the SADS-CoV infection in IPI-2I cells and Vero cells and spike-mediated syncytia formation in vero and HEK293T cells. There is no discussion as to why the authors used HEK293T cells and not IPI-2I cells in the spike-fusion experiment.
2. Does SADS-CoV infection induce syncytia formation at the site of infection in vivo like SARS-CoV2(PMID: 32085846, 33158808)? it would be great if authors could check the infected ileal tissue sections for the presence of syncytia by a staining spike or N protein antibody.
3. None of the tested siRNA gives a strong reduction of CH25H levels. It would be great if authors could show better KD either using the pool of all three siRNA or through CRISPR. this should improve KD efficiency and subsequently strong phenotype on virus replication.
Minor comments
Figure 1: Panels B, D and F. Authors should show all the data points in graphs and add the statistical test in the figure legends.
Figure 1E. Mention the name of the viral antigen in the IHC ileal tissue images.
Figure 2. Panels A, B, C and D authors should show all the data points in graphs and add the statistical test in the figure legends.
Figure 3. Panels B, D and E authors should show all the data points in graphs and add the statistical test in the figure legends.
Figure 4. Panels B, C, E and F authors should show all the data points in graphs and add the statistical test in the figure legends.
Figure 5. Panels B, C, E and F authors should show all the data points in graphs and add the statistical test in the figure legends.
Figure 6. Panel A, C, D and F authors should show all the data points in graphs and add the statistical test in the figure legends.
Figure 6, panel E, the quality of images is very poor, it would be great if authors could change them with better ones.
Figure 7, panel A-J, the authors should show all the data points in graphs and add the statistical test in the figure legends.
Comments on the Quality of English LanguageMinor editing of English language required.
Reviewer 2 Report
Comments and Suggestions for Authors
The authors descried that CH25H and its mutants lacking catalytic activity could inhibit SADS-CoV replication and that 25HC, which is the product of CH25H, also inhibited the replication. They confirmed that both CH25H and 25HC blocked S protein-mediated membrane fusion to inhibit the replication. Although the findings are of considerable interest, a number of points need clarifying and certain statements require further justification.
Major points:
1. The authors should mention why the mutant inhibited the replication.
2. Lines 362-367. The experiment is inappropriate to show direct inactivation with 25HC.
3. Lines 369-375. The experiment is also inappropriate.
Minor points:
1. Line 19. Porcine ileum epithelial cells not IPI-2I cells.
2. Line 67. Zika virus (ZIV).
3. Lines 87-90. This sentence should be modified, such as porcine ileum epithelial (IPI-2I) cells.
4. Table 1. Forward/reverse not 1st/2nd.
5. Line 138. H post-infection (hpi).
6. Line 151. The authors should indicate concentration of ethanol.
7. Line 156. The authors should indicate information of RiboBio.
8. Line s 181-182. The authors should indicate information of X-tremeGENE HP DNA transfection reagent.
9. Lines 196-197. These sentences should be revised.
10. Line 201 hpi not h post-infection.
11. Line 225. Delete IFN-stimulated gene ().
12. Line 248. An NC not a negative control siRNA (NC).
13. Lines 253-254. This sentence should be modified.
14. Lines 327-328. This sentence should be revised.
15. Line 329. PRRSV should be spelled out. Delete Zika virus ().
16. Lines 329-330. SARS-CoV, SARS-CoV-2,
17. Line 341. IFA.
18. Line 425. Delete plasmids.
19. Lines 440 and 443. Myc-SADS-CoV S-expressing plasmids.
30. Lines 455-456. This sentence should be revised.
31. Line 471. MBDMs not bone-marrow-derived macrophages.
32. Line 473. What are 3D4/21 cells?
33. Line 494. ZIV.
Comments on the Quality of English LanguageBe careful with use of abbreviations.
Round 2
Reviewer 2 Report
Comments and Suggestions for Authors
The manuscript has been improved. However, not a small number of points need to be revised. These are given below, although I cannot point out completely.
1. Scientific name should be in Italic.
2. Line 53. Spike (S) protein not Spike protein (S).
3. Line 58. SARS-CoV should be spelled out.
4. Line 72. Human immune deficiency virus (HIV).
5. Line 73. Reovirus is non-enveloped virus.
6. Line 86. IFN-stimulated gene (ISG). Porcine ileum epithelial cells not IPI-2I cells.
7. Lines 98-99. This sentence should be revised.
8. Lines 102, 106, and 107. The authors should indicate the source of these plasmids.
9. Lines 130-131. qRT-PCR not quantitative reverse transcription poly-130 merase chain reaction (qRT-PCR).
10. Line 137. The title should be modified.
11. Table 1. Sense not primer.
12. Line 149. Incubated not hybridized.
13.Line 154. virus not supernatants.
14. Line 159. Absolute alcohol must kill the cells.
15. Line 313. Mock-transfected not mock-infected.
16. Line 316. Delete and Vero E6 cells.
17. Line 336. Enzyme not cholesterol 25-hydroxylase.
18. Line 339. MERS should be spelled out.
19. Line 345. IPI-2I and Vero E6 cells.
20. Line 385-38. I wonder if the authors examined for vRNA not mRNA, because they incubated the cells with the virus only for 1 h at 4℃, what's more.
21. Line 392-393. Same as above, because the authors incubated the cells only for 2 h at 37℃. I also wonder if they detected the virus remained on the cell surface.
22. Fig. 8 The box plots should be in another panel.
23. Lines 473-474, These sentences should be combined.
24. Line 510. PRV should be spelled out.
25. Line 514. Delete sterol regulatory element binding protein 2 ( ).
26. Line 530. In a 25HC-dependent and 25HC-independent manners.
Comments on the Quality of English Language
Minor ~ moderate editing of English language required.
